# Fractionation of Transition Metals by Solvent Extraction and Precipitation from Tannic Acid-Acetic Acid Leachate as a Product of Lithium-Ion Battery Leaching

Erik Prasetyo [1,2,*], Corby Anderson [3], Arya Fitra Jaya [4], Widya Aryani Muryanta [1], Anton Sapto Handoko [1], Muhammad Amin [1], Muhammad Al Muttaqii [5] and Fathan Bahfie [1]

1   Research Center for Mining Technology, National Research and Innovation Agency, Jl. Ir. Sutami, km 15, Lampung Selatan 35361, Indonesia; widyaaryani95@gmail.com (W.A.M.); e_electrical@yahoo.com (A.S.H.); muhammad.amin@lipi.go.id (M.A.); fathanbahfie@gmail.com (F.B.)
2   Department of Chemical Engineering, Norwegian University of Science and Technology, Kjemi 4, Gløshaugen, N-7491 Trondheim, Norway
3   Department of Mining Engineering, Kroll Institute for Extractive Metallurgy, Colorado School of Mines, 1500 Illinois St., Hill Hall 337, Golden, CO 80401, USA; cgandersmines@gmail.com
4   Metallurgical Engineering, Institute of Technology Bandung, Labtek IVA, Lt. 2, Jl. Ganesha 10, Bandung 40132, Indonesia; aryafj98@gmail.com
5   Research Center for Chemistry, National Research and Innovation Agency, Gd. 452 Bldg, Jl Puspitek Serpong Gate, Muncul, Serpong, South Tangerang 15314, Indonesia; almuttaqiimuhammad@gmail.com
*   Correspondence: erik.prasetyo@ntnu.no

**Abstract:** Solvent extraction and precipitation schemes are applied to isolate copper, cobalt, manganese and nickel from leachate, produced from spent lithium-ion battery leaching using tannic acid-acetic acid as lixiviant. The metal separation and purification were developed based on a ketoxime (LIX® 84-I) and a phosphinic acid (Cyanex® 272) extraction system. Aside from the leachate's initial pH, which dictates the metal isolation flowsheet, other parameters affecting metal extraction rate, such as phase ratio, extractant concentration, and acid stripping will be evaluated. Copper was selectively removed from leachate at pH 3, using LIX® 84-I 10% *v/v* followed by cobalt and manganese co-extraction from the raffinate using Cyanex® 272 10% *v/v* at pH 5. After both metals were stripped using sulfuric acid 0.2 M, manganese was quantitatively precipitated out from the strip solution using potassium permanganate or sodium hypochlorite. Nickel was isolated using LIX® 84-I from raffinate at pH 5, producing a lithium- rich solution for further treatment. No third phase was formed during the extraction, and sulfuric acid was proved suitable for organic phase regeneration.

**Keywords:** solvent extraction; leaching; separation; purification

## 1. Introduction

The consumption rate of certain elements, i.e., lithium (Li), nickel (Ni), manganese (Mn) and cobalt (Co), sharply increased recently due to the surge of lithium-ion batteries (LIBs) production for electric vehicles (EVs). EV technology, including its energy storage device, has become intensively developed as an alternative to conventional combustion type vehicles, to reduce the carbon emissions associated with fossil fuel consumption [1]. Due to the uneven natural resource distribution of these elements [2], causing geopolitical criticality, and the massive volume of the battery that has already entered the end of life, recycling technology is required to secure the supply and minimize effect on the environment. The LIB recycling process can be classified into three categories: high-temperature process (pyrometallurgy), chemical process (hydrometallurgy) and direct recycling [3]. The hydrometallurgical process offers advantages in terms of low energy consumption, the ability to cope with low-grade raw materials, and the ability to separate and purify each element to yield marketable products.

Prior to recycling using chemical treatment, spent LIB is dismantled and pre-treated to separate each component. The cathode (Li metal oxide) and anode (graphite) are treated together and collectively called black mass. These components are the main targets in the recycling process due to their value, which comprises about 80% of the total LIB value [4]. In the hydrometallurgical process, the black mass is dissolved (leaching) using mineral acids such as sulfuric acid [5,6], nitric acid [7,8], hydrochloric acid [9,10], and phosphoric acid [11]. Organic acids, such as poly-carboxylic acid (citric acid [12], tartaric acid [13], lactic acid [14]), amino acids (glycine) [15] and polyphenols (tannic acid and tannin) [16] are used as less toxic and safer alternatives.

The target metals are dissolved in the leach liquor (leachate) during leaching, and separation and purification steps are required to recover these metals. These steps include solid-phase extraction (adsorption), precipitation, crystallization, solvent extraction (SX), or a combination of these. Adsorption is generally effective to recover target metal ions of which concentrations are within mg/L order [17]. In the case of precipitation and crystallization, the typical concentration is at the other end of the spectrum (tens of g/L) to reach saturated concentration in order to initiate the process. Precipitation and crystallization are considered less selective methods; hence the product is prone to impurities. Solvent extraction is regarded as the best separation and purification method, considering the typical metal concentration in leachate (g/L). A literature review on the treatment of leachate using solvent extraction reveals that most of the leachates were produced by black mass leaching using strong mineral acids, especially sulfuric acid (Table 1). This correlates with the extensive application of sulfuric acid as lixiviants in various battery recycling schemes, either in processes which are still in the development stage, such as the Aalto process [18], or in already commercialized processes, such as the Umicore and Batrec processes [2].

As mentioned in the previous paragraph, several organic acids have been intensively used as safer and environmentally friendly alternative lixiviants. Surprisingly, although optimum leaching conditions using these organic reagents have been established in numerous reports and publications, the separation and purification scheme using solvent extraction to isolate each element from organic leach liquor is exceptionally scarce. So far, the attempt to recover metals from organic leach liquor has been carried out by Punt et al. (2021), Table 1 [19]. Metal isolation from organic leach liquor might be more complicated than the isolation from inorganic media, such as sulfate and chloride, due to the possible formation of a strong complex between metal ions and organic acids. This reason might be why reports on solvent extraction from such media are so scarce. However, due to increasing interest in the LIB leaching scheme using organic acids, and considering the green chemistry principle, further investigation on metal isolation from organic leach liquor using solvent extraction is justified. Our previous study on black mass leaching using tannic acid-acetic acid as lixiviant [16] established the optimum leaching condition, producing leach liquor with a metal concentration suitable for solvent extraction. This paper reports the scheme to isolate transition metals, i.e., Cu, Mn, Co and Ni, using solvent extraction from leach liquor produced by black mass leaching using tannic acid-acetic acid. Commercial extractants, i.e., LIX® 84-I and Cyanex® 272, and kerosene as a diluent, will be tested for their performance in terms of extraction parameters, such as pH, organic-aqueous phase ratio, extractant concentration, and acid stripping using a batch method. Aside from solvent extraction, the precipitation method is also explored in the metal separation scheme.

**Table 1.** Previous investigation on elements separation and purification using solvent extraction from LIB leach liquor.

| No. | Leach Liquor | Elements | Extractant | Optimum Condition | Stripping Rate | Reference |
|---|---|---|---|---|---|---|
| 1 | Chloride | Co (1 g/L) | TOA 0.2 M in kerosene | O/A 1, HCl 6 M (99%) | 98% using Sulfuric acid 2 M | [20] |
| | | | Cyanex301 0.1 M in kerosene | O/A 1, pH 6–7 (95%) | 91% using Sulfuric acid 1 M | |
| 2 | Chloride | Co 2.9 g/L | Cyphos IL 102 0.2 M in toluene | O/A 1, HCl 8 M (94%) | 99% using HCl 0.05 M, A/O 1 | [21] |
| 3 | Sulfate | Co Cu (1.4 g/L) | Cyanex® 272 1 M Acorga M5640 10% | O/A 1, pH 5.5 (99%) O/A 1, pH 1 (96%) | | [22] |
| 4 | Sulfate | Co (12.3 g/L) Ni (28.5 g/L) Mn (14.65 g/L) | Versatic 10 50% Versatic 10 40% Versatic 10 50% | O/A 1, pH 6 (98.5%) O/A 1, pH 6 (99.2%) O/A 1, pH 6 (97%) | | [23] |
| 5 | sulfate | Mn 1.2 g/L Co 1 g/L Ni 0.8 g/L | TODGA in [C4mim][NTf2] [P66614][Cl] DES DecA:Lid | O/A 1, pH 3.3 (90%) H2SO4 9 M (93%) O/A 1, pH 4 (92%) | | [24] |
| 6 | Sulfate | Mn 10 g/L | DEHPA 0.5 M | O/A 1.25, pH 3.25 (70%) | Sulfuric acid 1 M | [25] |
| 7 | Sulfate | Mn 0.55 g/L | DEHPA 10% | O/A 1.2, pH 2.7 (84%) | | [26] |
| 8 | Sulfate | Mn 4.96 g/L | Na-DEHPA | O/A 2.33, pH 3 (88%) | | [27] |
| 9 | Sulfate | Co 14 g/L | Cyanex® 272 1 M, TOA 5% | O/A 1.3, pH 6.8–7.1 (99%) | pH 5–6, A/O 0.67 | [28] |
| | | Ni 0.5 g/L | Cyanex® 272 1 M, TOA 5% | O/A 1.3, pH 6.8–7.1 (>90%) | | |
| 10 | Citrate | Mn | DEHPA 12% | O/A 2, pH 2.5 (99%) | Citric acid 1.5 M | [19] |

## 2. Materials and Methods

### 2.1. Chemicals and Apparatus

Extractant LIX 84-I (active component 2-hydroxy-5-nonylaceto-phenone oxime) and Cyanex® 272 (bis(2,2,4 trimethylpentyl)phosphinic acid) were procured from PT BASF Distribution Indonesia and Cytec USA, respectively. Kerosene (low odor) as diluent was purchased from Sigma-Aldrich, St. Louis, MO, USA. Sodium hydroxide, sulfuric acid, potassium permanganate, oxalic acid and glacial acetic acid were obtained from Merck, Darmstadt, Germany, while tannic acid was obtained from Bean Town Chemical, Hudson, NH, USA. All reagents were used as received. MilliQ water (Merck Millipore, Burlington, MA, USA) was used throughout the experiments. Leach liquor was prepared by leaching LIB black mass with tannic acid–acetic acid mixture as lixiviants. LIB black mass was obtained from spent LIB 18650 type. After discharge and manual dismantling, the cathode and anode components were ball milled and sieved. Black mass powder (size less than 53 µm) was leached at the optimum conditions outlined in the previous study [16] (tannic acid 20 g/L-acetic acid 1 M, pulp density 20 g/L, stirring speed 250 m, leaching temperature 80 °C, 6 h). The concentration of each metal in leach liquor (determined using ICP-OES) is listed in Table 2.

**Table 2.** Metal content in leach liquor used in the solvent extraction and precipitation study.

| Element | Cu | Co | Mn | Ni | Li |
|---|---|---|---|---|---|
| Concentration (g/L) | 0.25 | 3.3 | 0.14 | 0.13 | 0.6 |

The pH of leach liquor produced from the leaching was 3. For the study on the pH effect on solvent extraction and precipitation rate, the pH of leach liquor was adjusted using sodium hydroxide 10 M or concentrated sulfuric acid. The pH was monitored using a pH meter (Oakton 45, Vernon Hills, IL, USA). The metal concentration in the aqueous phase before and after extraction or stripping, and before and after precipitation was determined using ICP-OES (Analytik Jena, Plasma Quant 9000 Elite, Jena, Germany).

### 2.2. Extraction and Stripping Test

The extraction test was carried out by the batch method, by introducing, typically, 10 mL leach liquor and 10 mL organic phase (mixture of kerosene with either LIX® 84-I or Cyanex® 272) into a 125 mL sealed conical flask. The mixture was homogenized by magnetic stirring (600 rpm, 15 min). After extraction, both organic and aqueous phases were separated. No third phase was observed, and 10 min was sufficient for complete phase disengagement. Centrifugation (4000 rpm, 5 min) was performed to ensure separation of the phases. The aqueous phase was then sampled, and the metal concentration was determined using ICP-OES. In the stripping test, typically, 10 mL loaded organic was homogenized with 10 mL inorganic acid in a 125 mL conical flask using a magnetic stirrer (600 rpm, 15 min). After phase disengagement, the strip solution was sampled, and the metal content was determined using ICP-OES. Extraction recovery ($R$, %), distribution coefficient ($D$) and log separation factor ($\beta$) were calculated using Equations (1)–(3) while stripping recovery ($S$, %) was evaluated using Equation (4). All extraction and stripping tests were done in duplicates. The effect of organic (O)-aqueous (A) volume ratio on metal extraction and metal stripping recovery was studied by the batch method, which represented the cross-current extraction scheme in a McCabe-Thiele plot.

$$R = \frac{C_o - C_E}{C_o} \times 100\% \tag{1}$$

$$D = \frac{C_o - C_E}{C_E} \times \frac{V_{aq}}{V_{org}} \tag{2}$$

$$\beta = |\log D_1 - \log D_2| \tag{3}$$

$$S = \frac{C_s V_s}{C_{org} V_{org}} \times 100\% \tag{4}$$

where:

$C_o$: Metal concentration in aqueous phase before extraction (mg/L);
$C_E$: Metal concentration in aqueous phase after extraction (mg/L);
$V_{aq}$: Aqueous phase volume (mL);
$V_{org}$: Organic phase volume (mL);
$D_1$: Distribution coefficient of metal 1;
$D_2$: Distribution coefficient of metal 2;
$C_s$: Metal concentration in strip solution (mg/L);
$C_{org}$: Metal concentration in loaded organic phase (mg/L);
$V_s$: Volume of strip solution (mL).

### 2.3. Precipitation Test

The precipitation test was performed to separate Co-Mn from strip solution. Three kinds of precipitants were tested. Oxalic acid was used to precipitate Co out from the strip solution, while potassium permanganate and sodium hypochlorite were used for selective Mn precipitation. Typically, 25 mL of strip solution, containing both Co and Mn, were mixed with precipitant, in which the precipitant/metal molar ratio varied between 0.13–5. The precipitation occurred spontaneously, and to ensure complete precipitation, the mixture was let to stand for 12 h at room temperature. After that, the filtrate was sampled, and the metal concentration was determined using ICP-OES. Precipitation rate ($P$, %) and precipitation selectivity ($\theta$) were used to evaluate the efficacy of the precipitation scheme in Co-Mn separation, according to Equations (5) and (6), respectively. All precipitation tests were carried out in duplicates.

$$P = \frac{C_s V_s - C_f (V_s + V_P)}{C_s V_s} \times 100\% \tag{5}$$

$$\theta = \frac{C_{f1}}{C_{f2}} \tag{6}$$

where:

$C_s$: Metal concentration in strip solution (mg/L);
$V_s$: Volume of strip solution (mL);
$C_f$: Metal concentration in filtrate after precipitation (mg/L);
$V_p$: Volume precipitant added (mL);
$C_{f1}$: Metal 1 concentration in filtrate after precipitation (mg/L);
$C_{f2}$: Metal 2 concentration in filtrate after precipitation (mg/L).

## 3. Results and Discussions

### 3.1. pH Effect on Cu, Mn, Co, and Ni Extraction

The effect of pH of leach liquor on metal extraction was firstly carried out to determine the possible separation scheme for each transition metal, which was investigated within range 1–7. The constant variables in the study included organic/aqueous (O/A) volume ratio 1, extractant concentration 20% *v/v*, and equilibrium time 15 min at room temperature. The results of the studies are depicted in Figure 1. Cyanex® 272 effectively transferred Cu, Co and Mn (Figure 1a) while rejecting Ni (pH 1–4) and Li (pH 1–5). The maximum extraction rate for Ni and Li were 24% and 27% at pH 7, respectively. Equilibrium $pH_{50}$ for the three metals (Mn, Co and Cu) were estimated to be 2.2, 2.3 and 2.5 (Figure S1), respectively. These results were lower compared to $pH_{50}$ values obtained from sulfate media ($pH_{50}$ Cu 3, $pH_{50}$ Co 3.75 and $pH_{50}$ Mn 3.5) reported by Solvay [29]. Previous reports on solvent extraction of transition metals from sulfate media revealed that the presence of acetate ions would decrease the $pH_{50}$, making the metal extraction possible in more acidic conditions [30]. Increasing metal transfer at lower pH was hypothetically caused by the ability of acetate to buffer the pH during the extraction, and not by the role of acetate in complex (metal-ligand) formation during the extraction. However, based on modelling using Hydra and Medusa, Cu, Co, Mn and Ni form acetate complex in a weak acidic to neutral pH range (Figure S2). In the case of Co and Mn, both metals produce cationic species $Co(CH_3COO)^+$ and $Mn(CH_3COO)^+$, respectively, while Ni and Cu form neutral species $Ni(CH_3COO)_2$, and anionic species $Cu(CH_3COO)_3^-$, respectively. This explains why Cyanex® 272 (acidic extractant) had stronger affinity to Co and Mn, due to the positive charge of both metal species, compared to Cu and Ni, which as complexes are less charged.

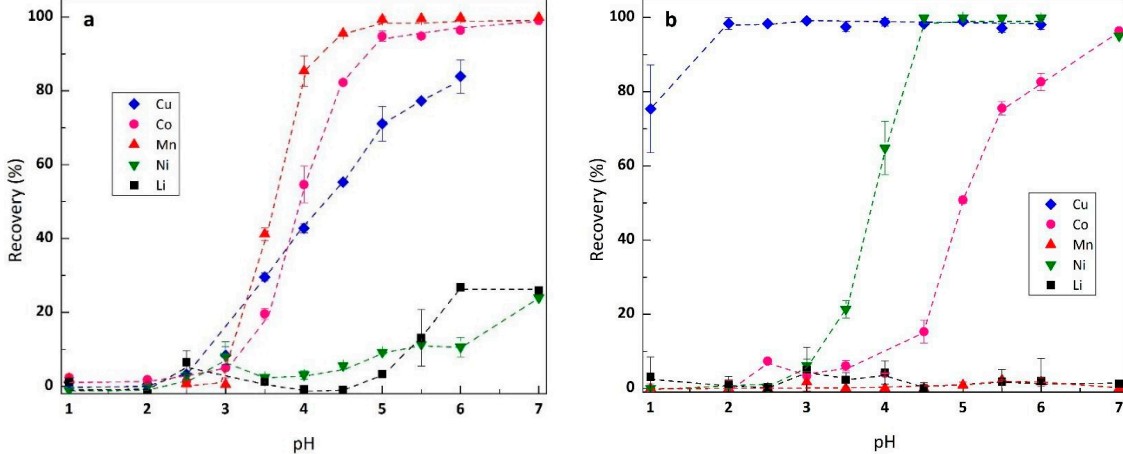

**Figure 1.** The extraction rate of the metal ion by (**a**). Cyanex® 272 10% *v/v* and (**b**). LIX® 84-I 10% *v/v* according to the initial pH of leach liquor.

In the case of LIX® 84-I (Figure 1b), the extractant was applicable for Cu, Co and Ni. Cu extraction using LIX® 84-I was effective at pH higher than 2 (recovery >99%), while for

Co and Ni, the pH$_{50}$ was estimated to be 5 and 3.5, respectively. In the case of Mn and Li, both metals were rejected by LIX® 84-I within the studied pH range. The effect of acetate in Cu and Ni extraction using oxime extractant was positive, increasing the metal transfer from the aqueous to the organic phase. Characterization of the organic phase loaded with Cu and Ni, using FT-IR, shows no significant differences between organics produced from acetate and non-acetate media (Figure S3). This indicates that acetate has no significant role in extractable complex formation during metal transfer. Instead, the increasing transfer rate in acetate media was due to the buffering effect of acetate on the pH of the aqueous phase.

Based on pH studies, sequential isolation of transition metals from tannic acid-acetic acid leach liquor is proposed as follows:

1. Cu is first isolated from leach liquor (initial pH 3) using LIX® 84-I, leaving Co, Mn, Ni and Li in the raffinate. The distribution coefficient of Cu using LIX® 84-I at this pH was 80.8, and log separation factors with Co, Mn, Ni and Li were 3.00, 3.64, 3.07 and 4.15, respectively.
2. Co and Mn are co-extracted using Cyanex® 272 from the raffinate after the pH of raffinate is increased to 5. Only Ni and Li were left in the raffinate after Co-Mn co-extraction. The distribution coefficient of Co was 19.76 and Mn was 89.1 at pH 5, while log separation factors were 2.27 (Co- Ni), 3.75 (Co-Li), 2.92 (Mn-Ni) and 4.41 (Mn-Li).
3. Ni is separated from Li using LIX® 84-I at pH 5, leaving Li in the final raffinate (DNi 732, βNi-Li 5.25).

Precipitation is proposed to separate Co and Mn in strip solution. Co is precipitated out from strip solution as oxalate, while Mn is selectively precipitated as MnO$_2$ using hypochlorite or permanganate to separate Mn from Co.

### 3.2. Cu Recovery from Leachate

### 3.2.1. Cu Extraction Using LIX® 84-I

The first step in transition metals fractionation from leachate is Cu isolation using LIX® 84-I. Figure 2 shows the effect of LIX® 84-I concentration in the organic phase on Cu extraction and other metals. The concentration varied between 5 and 30% *v/v*, while initial aqueous pH, A/O volume ratio, equilibrium time, and reaction temperature were set constant at 3, 1, 15 min, and room temperature. It is demonstrated that 5% LIX® 84-I is sufficient to quantitatively remove Cu from leach liquor (>99%). Besides, increasing LIX® 84-I concentration did not significantly extract other metals, confirming Cu extraction's selectivity at the proposed condition. Isotherm studies on the A/O ratio effect (1–7) on Cu recovery (Figure 3) resulted in a maximum loading capacity of LIX® 84-I 10% *v/v* 1.2 g/L Cu (Cu in feed solution 0.23 g/L). The loading capacity might be higher in the case of using a higher A/O value or using a continuous test, instead of a batch test. Table 3 lists the number of theoretical stages required for complete Cu isolation from feed solution at different A/O ratios. Due to the high transfer rate, at A/O ratio 7, the transfer rate of Cu was 81%, which theoretically required two stages for complete Cu extraction.

**Table 3.** Cu extraction rate and theoretical stages required for complete extraction at different A/O ratios (LIX® 84-I 10% *v/v*, feed solution metal concentration as in Table 2, pH 3).

| A/O Ratio *v/v* | Cu Rec (%) | Number of Theoretical Stages |
|---|---|---|
| 1 | 99.34 ± 0.07 | 1 |
| 2 | 97.83 ± 2.05 | 2 |
| 3 | 96.04 ± 1.72 | 2 |
| 4 | 96.99 ± 2.87 | 2 |
| 5 | 98.10 ± 1.77 | 2 |
| 6 | 85.89 ± 0.05 | 2 |
| 7 | 81.41 ± 8.86 | 2 |

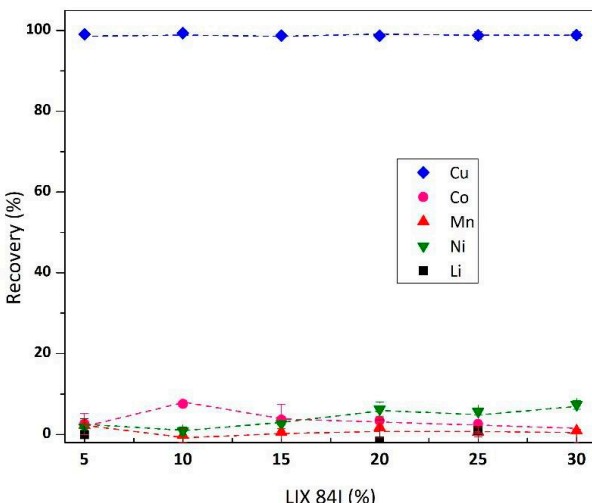

**Figure 2.** Effect of LIX® 84-I concentration (*v/v*) on Cu isolation from leachate (pH 3, metal concentration in feed Cu 0.23 g/L, Co 3.54 g/L, Mn 0.16 g/L, Ni 0.14 g/L and Li 0.62 g/L).

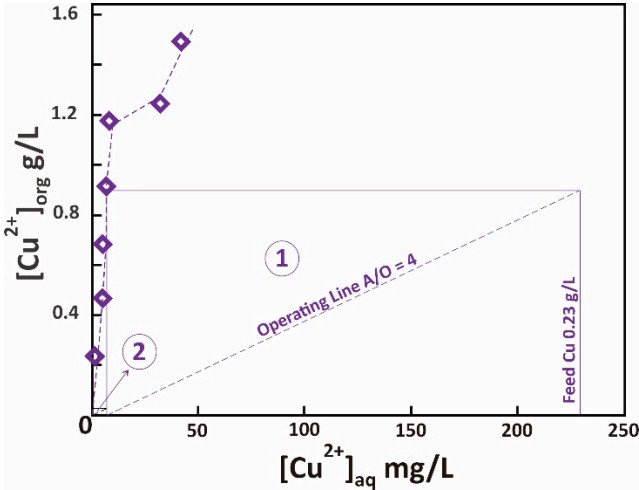

**Figure 3.** Extraction isotherm of Cu using LIX® 84-I 10% *v/v*, pH 3, McCabe-Thiele plot is shown for A/O ratio 4.

### 3.2.2. Stripping of Cu

Cu was stripped from the loaded organic (LIX® 84-I 10% *v/v*) using the most common mineral acids, i.e., sulfuric acid, nitric acid and hydrochloric acid. The Loaded organic (Cu content 4.8 g/L) was stripped at O/A volume ratio 1, room temperature and 15 min equilibration, while acid concentration varied from 0.1 to 5 M. Table 4 shows the effect of acid concentration on the stripping rate of Cu from loaded organic. There is no significant effect on the kind of acid used. For single-stage stripping, the minimum acid concentration required for complete Cu stripping was 1 M. Increasing sulfuric acid slightly decreased the Cu stripping recovery, probably caused by possible Cu precipitation in strip solution due to increasing sulfate concentration.

**Table 4.** Cu stripping recovery according to mineral acid type and acid concentration (O/A 1, feed organic LIX® 84-I 10% *v/v*, Cu 4.8 g/L).

| Acid Concentration (M) | Cu Stripping Recovery (%) | | |
|---|---|---|---|
| | Sulfuric Acid | Nitric Acid | Hydrochloric Acid |
| 0.1 | 61.05 ± 2.90 | 35.62 ± 2.75 | 33.93 ± 0.47 |
| 0.5 | 84.76 ± 4.99 | 72.19 ± 1.63 | 70.70 ± 0.14 |
| 1 | 97.11 ± 2.77 | 94.35 ± 6.33 | 95.75 ± 2.75 |
| 2 | 95.38 ± 0.40 | 97.23 ± 0.78 | 98.46 ± 0.35 |
| 5 | 89.33 ± 0.91 | 94.79 ± 0.42 | 100.75 ± 1.56 |

### 3.3. Simultaneous Co-Mn Recovery from Raffinate

### 3.3.1. Co-Mn co-Extraction Using Cyanex® 272

Co and Mn were co-extracted from raffinate using Cyanex® 272 after the pH of raffinate was adjusted to 5 using sodium hydroxide. Figure 4 demonstrates the effect of Cyanex® 272 concentration (5–30%) on the Co and Mn extraction rate (constant variable feed solution Co 3.3 g/L, Mn 0.16 g/L, Ni 0.13 g/L, Li 0.62 g/L, A/O ratio 1, 15 min equilibrium at room temperature). The figure shows that the minimum Cyanex® 272 concentration for Co-Mn co-extraction is 20%. No significant Ni and Li recovery were observed as Cyanex® 272 concentration increased. Extraction isotherm study discerned the effect of A/O ratio (1–7.5) to the Co-Mn co-extraction (Figure 5), resulting in maximum Co and Mn loading capacity of Cyanex® 272 to be 3.5 g/L Co and 0.53 g/L Mn. Lower loading capacity of Cyanex® 272 by Mn was due to stronger affinity and far higher concentration of Co in the feed solution. The McCabe-Thiele plot in Figure 5 gives the number of theoretical stages required for complete Co-Mn co-extraction using Cyanex® 272 10% *v/v*, which is resumed in Table 5.

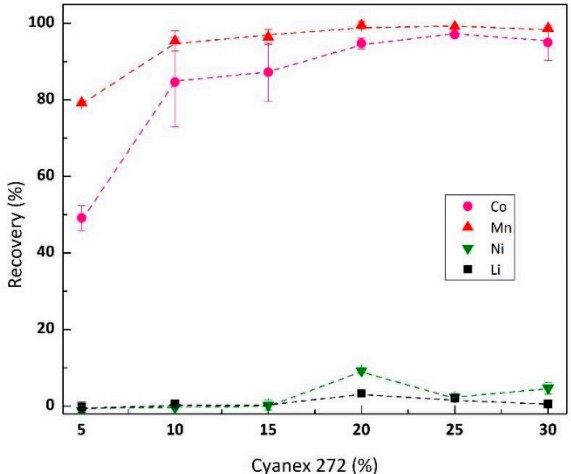

**Figure 4.** Effect of Cyanex® 272 concentration (*v/v*) on Co-Mn co-extraction from raffinate (pH 5, metal concentration in feed Co 3.3 g/L, Mn 0.16 g/L, Ni 0.14 g/L and Li 0.62 g/L).

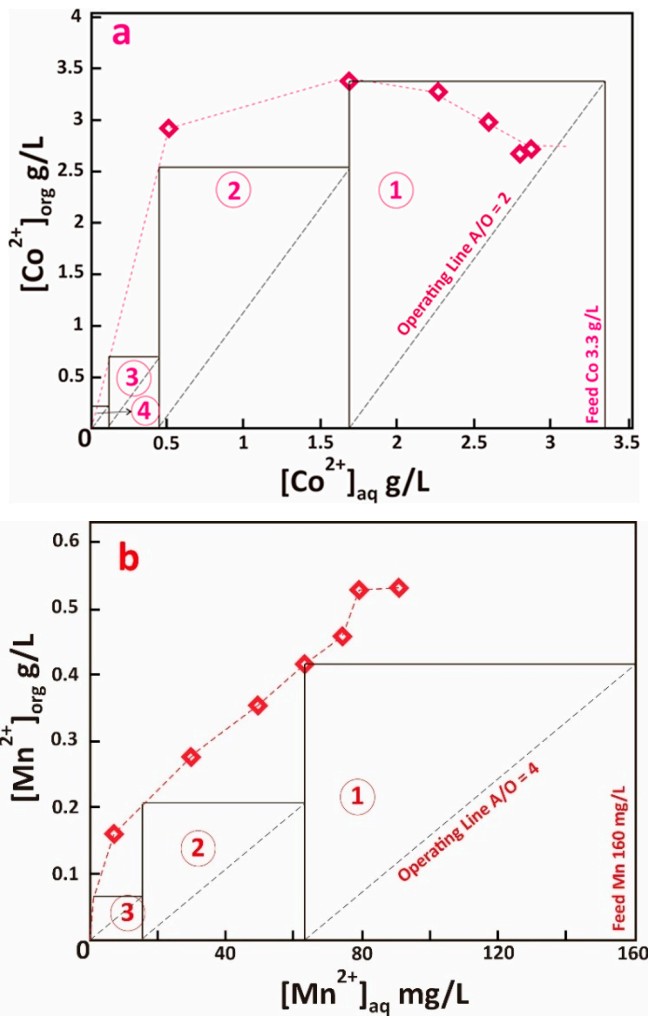

**Figure 5.** Extraction isotherm of (**a**). Co and (**b**). Mn using Cyanex® 272 10% *v/v*, pH 5, McCabe-Thiele plot is shown for A/O ratio 2 (Co) and 4 (Mn).

**Table 5.** Co-Mn extraction rate and theoretical stages required for complete extraction at different A/O ratio (Cyanex® 272 10% *v/v*, feed solution Co 3.3 g/L, Mn 0.16 g/L, pH 5).

| A/O Ratio *v/v* | Co | | Mn | |
|---|---|---|---|---|
| | Rec (%) | Number of Theoretical Stages | Rec (%) | Number of Theoretical Stages |
| 1 | 84.67 ± 11.72 | 2 | 95.44 ± 2.63 | 2 |
| 2 | 48.86 ± 5.07 | 4 | 81.37 ± 2.82 | 3 |
| 3.2 | 31.35 ± 0.90 | 4 | 69.30 ± 0.69 | 3 |
| 4.2 | 21.35 ± 1.63 | 5 | 60.83 ± 0.62 | 3 |
| 5.3 | 15.31 ± 3.86 | 7 | 53.90 ± 1.62 | 4 |
| 6.4 | 12.93 ± 0.68 | 8 | 51.03 ± 2.01 | 4 |
| 7.5 | 8.05 ± 0.73 | 10 | 43.79 ± 0.77 | 5 |

### 3.3.2. Co-Mn Stripping Using Sulfuric Acid

Stripping the loaded organic (Cyanex® 272 10% *v/v*, Co 3.3 g/L and Mn 0.17 g/L) was performed by 10 mL loaded organic with 10 mL sulfuric acid (0.1–2 M). The results in Figure 6 demonstrate that a low concentration of sulfuric acid 0.1 M is sufficient for quantitative Co and Mn stripping (Co 95.1%, Mn 102.2%). The trend of stripping rate in Figure 6 tends to decrease slightly as the sulfuric acid concentration rises. This was

probably due to increasing sulfate in the stripping solution, which depressed the solubility of $CoSO_4$ and $MnSO_4$ (effect of similar ion).

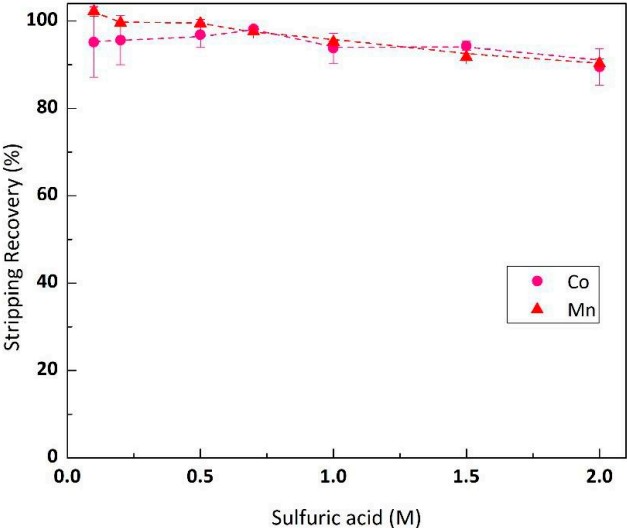

**Figure 6.** Effect of sulfuric acid concentration on Co-Mn stripping rate from loaded organic (O/A ratio 1, organic feed Cyanex® 272 10% *v/v*, Co 3.3 g/L, Ni 0.17 g/L).

The results of stripping isotherm of loaded organic containing Co-Mn using sulfuric acid 0.2 M, in which the A/O ratio varied between 2–5, are depicted in Figure 7. The maximum stripping capacity of sulfuric acid 0.2 M for Co and Mn were 11 g/L and 0.5 g/L, respectively. The number of theoretical stages required for complete stripping of Co and Mn using sulfuric acid 0.2 M at different O/A ratios, based on the McCabe-Thiele plot, are summarized in Table 6.

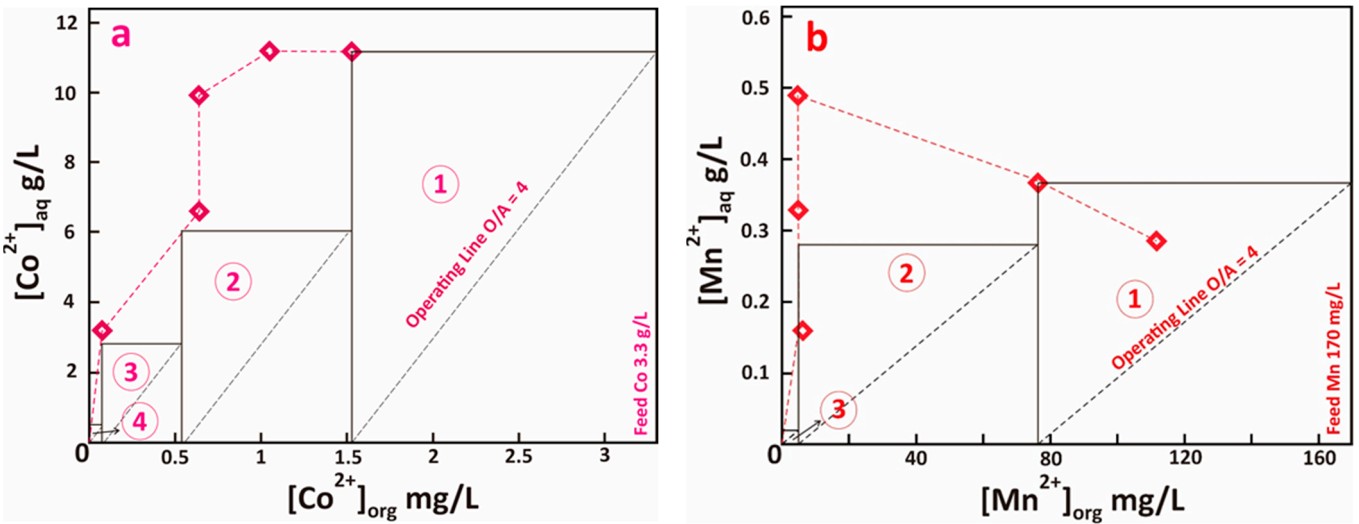

**Figure 7.** Stripping isotherm of (**a**). Co and (**b**). Mn from loaded organic (feed Co 3.3 g/L, Mn 0.17 g/L) using sulfuric acid 0.2 M, McCabe-Thiele plot is shown for O/A ratio 5 (Co) and 4 (Mn).

**Table 6.** Co-Mn stripping rate and theoretical stages required for complete stripping at different O/A ratios using sulfuric acid 0.2 M (feed organic Cyanex® 272 10% *v/v*, Co 3.2 g/L, Mn 0.16 g/L).

| O/A Ratio *v/v* | Co | | Mn | |
|---|---|---|---|---|
| | Rec (%) | Number of Theoretical Stages | Rec (%) | Number of Theoretical Stages |
| 1 | 95.62 ± 3.95 | 2 | 99.54 ± 0.06 | 2 |
| 2 | 81.02 ± 0.87 | 2 | 97.49 ± 0.68 | 2 |
| 3 | 81.15 ± 3.29 | 2 | 97.58 ± 0.23 | 2 |
| 4 | 68.89 ± 1.42 | 3 | 54.86 ± 0.53 | 3 |
| 5 | 54.72 ± 1.29 | 3 | 33.91 ± 0.18 | 3 |

*3.4. Mn-Co Separation from Strip Solution*

3.4.1. Co Selective Precipitation Using Oxalic Acid

Co and Mn were co-stripped from loaded organic using sulfuric acid 0.2 M. The strip solution obtained contained a mixture of Co (12 g/L) and Mn (0.5 g/L), in which final pH was 1. There were three precipitants tested for selective precipitation of Co or Mn, i.e., oxalic acid for Co precipitation, sodium hypochlorite and potassium permanganate for Mn precipitation. Oxalic acid is the common reagent used in Co precipitation and was chosen based on the ratio between solubility between two metal oxalic salts being higher than $10^7$ (Co oxalate solubility $2.6972 \times 10^{-8}$ g/L, Mn oxalate dihydrate 0.33 g/L). The independent variables in oxalate precipitation were pH (1–5) and oxalate/Co mol ratio. The results in Table 7 show no apparent effect of pH on Co precipitation. Based on species distribution modelling using Hydra and Medusa (Supplementary Figure S4), Co precipitation should not be affected by pH because at pH 0–10, Co exists as oxalate complex. In the case of Mn, Mn oxalate formation occurs at pH 1–10 (Figure S4b). The maximum Mn precipitation rate happened at pH 3 in accordance with modelling in Figure S4b, in which at pH 3, the percentage of Mn-oxalate species reaches a maximum.

**Table 7.** The precipitation rate and selectivity of Co and Mn using oxalic acid based on mol ratio of Oxalate/Co and initial pH of strip solution.

| | Precipitation Rate (%) | | Precipitation Selectivity |
|---|---|---|---|
| | Co | Mn | Mn% in Co Precipitate |
| pH | Oxalate/Co mol ratio 0.5 | | |
| 1 | 56.58 ± 0.17 | 1.86 ± 0.44 | 0.13 ± 0.03 |
| 2 | 64.40 ± 0.22 | 7.73 ± 0.11 | 0.33 ± 0.00 |
| 3 | 49.40 ± 0.02 | 0.72 ± 0.65 | 0.05 ± 0.05 |
| 4 | 45.38 ± 0.16 | 8.25 ± 1.85 | 0.74 ± 0.16 |
| 5 | 50.03 ± 0.09 | 2.09 ± 1.77 | 0.16 ± 0.13 |
| pH | Oxalate/Co mol ratio 0.75 | | |
| 1 | 71.64 ± 0.10 | 11.13 ± 0.36 | 0.58 ± 0.02 |
| 2 | 72.28 ± 0.12 | 14.95 ± 0.13 | 0.80 ± 0.06 |
| 3 | 72.75 ± 0.17 | 12.89 ± 0.01 | 0.64 ± 0.00 |
| 4 | 72.39 ± 0.93 | 11.26 ± 0.31 | 0.58 ± 0.02 |
| 5 | 73.36 ± 0.19 | 15.67 ± 1.21 | 0.82 ± 0.07 |
| pH | Oxalate/Co mol ratio 1 | | |
| 1 | 90.09 ± 0.07 | 15.90 ± 5.32 | 0.68 ± 0.23 |
| 2 | 90.65 ± 0.22 | 15.05 ± 1.40 | 0.65 ± 0.06 |
| 3 | 91.57 ± 0.19 | 17.65 ± 6.83 | 0.76 ± 0.29 |
| 4 | 91.17 ± 0.26 | 20.55 ± 1.91 | 0.88 ± 0.08 |
| 5 | 90.62 ± 0.60 | 17.80 ± 0.97 | 0.75 ± 0.04 |
| pH | Oxalate/Co mol ratio 1.2 | | |
| 1 | 91.12 ± 1.98 | 29.00 ± 1.94 | 1.25 ± 0.06 |
| 2 | 93.51 ± 0.11 | 30.16 ± 1.68 | 1.27 ± 0.07 |
| 3 | 95.63 ± 0.11 | 54.49 ± 0.48 | 2.22 ± 0.02 |
| 4 | 93.15 ± 0.12 | 31.46 ± 1.44 | 1.36 ± 0.06 |
| 5 | 92.84 ± 0.50 | 30.29 ± 0.47 | 1.31 ± 0.01 |

In the case of mol ratio variables, higher mol ratio promotes both Co and Mn precipitation rates. At stoichiometric ratio (oxalate/Co ratio 1), Co was not completely precipitated out of the stripping solution (Co precipitation rate 91%), while at excess ratio (1.2), the precipitation rate of Co only slightly improved. After precipitation, Mn concentration in the supernatant was not much varied in different pH tests, but was rather affected by oxalate/Co mol ratio. At mol ratio 0.5, the Mn concentration in filtrate after precipitation was ≈370 mg/L. The value decreased to ≈200 mg at the mol ratio of 1.2. These values are far higher than Mn solubility as oxalate reported in the literature (126 mg/L). Oxalate precipitation selectivity is reviewed by calculating the percentage of Mn in the precipitate (Table 7) using mass balance calculation. Increasing mol ratio caused more Mn impurities in Co-oxalate precipitate. The highest purity was obtained at oxalate/Co mol ratio 0.5 (Mn% impurity < 0.74%), while at excess addition of oxalate (mol ratio 1.2), the Mn impurity could reach 2.22%. The theoretically highest percentage of Mn impurity is 4% (all Mn and Co are precipitated). This signifies the efficacy of Co separation from Mn by oxalate precipitation.

### 3.4.2. Oxidative Mn Precipitation Using Sodium Hypochlorite

Mn in sulfate strip solution was precipitated using sodium hypochlorite as an oxidant. Table 8 summarizes the effect of hypochlorite/Mn mol ratio and strip solution pH on the precipitation rate of Mn and Co. Results in Table 8 confirm that hypochlorite is a selective precipitant in Co-Mn separation from strip solution, since the reduction potential of hypochlorite (Equations (7)–(9)) is far below the potential reduction $Co^{3+}/Co^{2+}$ (Equation (10)). Most of the Mn was removed from the strip solution by the excess of hypochlorite (mol ratio 3 and 5), which resulted in Co concentration in strip solution being about $10^3$ times higher than Mn (hypochlorite/Mn mol ratio 3) and about $10^5$ times at hypochlorite/Mn mol ratio 5.

$$ClO^-_{(aq)} + H_2O(l) + 2e \rightarrow Cl^-_{aq} + 2OH^-_{(aq)} \quad E^o = 810 \text{ mV} \tag{7}$$

$$HClO_{(aq)} + H^+_{(aq)} + e \rightarrow \frac{1}{2}Cl_{2(g)} + H_2O_{(l)} \quad E^o = 1611 \text{ mV} \tag{8}$$

$$HClO_{(aq)} + H^+_{(aq)} + 2e \rightarrow Cl^-_{(aq)} + H_2O_{(l)} \quad E^o = 1482 \text{ mV} \tag{9}$$

$$Co^{3+}_{(s)} + e \rightarrow Co^{2+}_{(aq)} \quad E^o = 1920 \text{ mV} \tag{10}$$

**Table 8.** The precipitation rate and selectivity of Co and Mn using sodium hypochlorite based on mol ratio of hypochlorite/Mn and initial pH of the strip solution.

| | Precipitation Rate (%) | | Precipitation Selectivity |
|---|---|---|---|
| | **Co** | **Mn** | **Log [Co/Mn] in Solution** |
| pH | Hypochlorite/Mn mol ratio 2 | | |
| 3 | −1.11 ± 0.68 | 63.21 ± 0.95 | 1.87 |
| 5 | 2.61 ± 0.61 | 68.26 ± 2.62 | 1.91 |
| 7 | −3.15 ± 0.16 | 66.78 ± 1.85 | 1.92 |
| pH | Hypochlorite/Mn mol ratio 3 | | |
| 3 | −3.95 ± 0.49 | 92.37 ± 3.63 | 2.60 |
| 5 | −2.05 ± 1.03 | 94.57 ± 0.88 | 2.69 |
| 7 | 0.96 ± 0.42 | 92.61 ± 5.41 | 2.70 |
| pH | Hypochlorite/Mn mol ratio 5 | | |
| 3 | −1.64 ± 0.22 | 99.96 ± 0.02 | 4.83 |
| 5 | 1.38 ± 1.33 | 100.04 ± 0.02 | 4.93 |
| 7 | 4.33 ± 0.75 | 100.02 ± 0.00 | 4.92 |

The oxidation of Mn by hypochlorite might occur according to several possible Equations (11)–(13):

$$Mn^{2+}_{(aq)} + 2ClO^-_{(aq)} \rightarrow MnO_2(s) + Cl_{2(g)} \qquad E^o = 387 \text{ mV} \qquad (11)$$

$$Mn^{2+}_{(aq)} + Cl_{2(g)} + 2H_2O_{(l)} \rightarrow MnO_{2(s)} + 4H^+_{(aq)} + 2Cl^-_{(aq)} \quad E^o = 134 \text{ mV} \qquad (12)$$

$$Mn^{2+}_{(aq)} + ClO^-_{(aq)} + H_2O_{(l)} \rightarrow MnO_{2(s)} + 2H^+_{(aq)} + Cl^-_{(aq)} \quad E^o = 258 \text{ mV} \qquad (13)$$

Equations (11) and (12) theoretically occurs only in the presence of excess acid. Since the precipitation condition was set at weakly acidic (pH 3) until neutral condition, the oxidation of Mn proceeded according to the Equation (13). Based on Table 8, the precipitation rate of Mn was higher in more alkaline conditions. The presence of $OH^-$ in solution neutralized protons, as an oxidation product (reaction 13), which pushed the reaction toward the product side. Increasing pH, however, also induced the precipitation of Co. At pH 7, in the presence of hypochlorite at mol ratios 3 and 5, the precipitation rate of Co is 1% and 4.3%, respectively. This was probably caused by Co precipitation as hydroxide. Calculation using Co concentration in strip solution and solubility product of $Co(OH)_2$, $1.60 \times 10^{-15}$, determined that Co started to precipitate as hydroxide at pH 6.97. There was also the possibility of Co oxidation by hypochlorite since Co oxidation potential decreases substantially at more alkaline conditions (Equation (14)).

$$Co(OH)_{3(s)} + e \rightarrow Co(OH)_{2(s)} + OH^-_{(aq)} \quad E^o = 170 \text{ mV} \qquad (14)$$

### 3.4.3. Oxidative Mn Precipitation Using Potassium Permanganate

Oxidative precipitation of Mn using potassium permanganate was performed at different permanganate/Mn molar ratios (0.17–1.67) and different pH (3.5 and 7). The oxidation of Mn by permanganate occurs according to Equation (15).

$$3Mn^{2+}_{(aq)} + 2MnO^-_{4(aq)} + 2H_2O_{(l)} \rightarrow 5MnO_{2(s)} + 4H^+_{(aq)} \quad E^o = 255 \text{ mV} \qquad (15)$$

Stoichiometrically, permanganate/Mn mol ratio 0.67 is sufficient for complete $Mn^{2+}$ removal from strip solution. Results in Table 9 show that the maximum precipitation rate for Mn at this mol ratio was 91%. A higher mol ratio resulted in a lower precipitation rate, due to excess permanganate added to the strip solution. Apparently, pH did not affect the precipitation rate and precipitation selectivity as in the case of hypochlorite precipitation. Oxidative precipitation is considered selective, since the oxidation potential of permanganate ($V = 1679$ mV) is still lower than the reduction potential of $Co^{3+}/Co^{2+}$ couple ($V = 1920$ mV). However, Co precipitation was quite significant in the presence of excess permanganate (Table 9). At optimum mol ratio (2/3), the precipitation rate could reach 5% at weak acidic-neutral conditions, and 12% at acidic conditions. It is still unclear why the precipitation of Co was significant under these conditions.

**Table 9.** The precipitation rate and selectivity of Co and Mn using potassium permanganate based on mol ratio of permanganate/Mn and initial pH of the strip solution.

| | Precipitation Rate (%) | | Precipitation Selectivity |
|---|---|---|---|
| | **Co** | **Mn** | **Log [Co/Mn] in Solution** |
| pH | Permanganate/Mn$^{2+}$ mol ratio 1/6 | | |
| 3 | $1.20 \pm 0.02$ | $24.08 \pm 0.62$ | 1.51 |
| 5 | $-1.99 \pm 2.01$ | $24.24 \pm 1.04$ | 1.52 |
| 7 | $-1.61 \pm 0.08$ | $23.67 \pm 1.43$ | 1.52 |
| pH | Permanganate/Mn$^{2+}$ mol ratio 1/3 | | |
| 3 | $1.42 \pm 3.47$ | $34.18 \pm 17.91$ | 1.58 |
| 5 | $2.61 \pm 0.78$ | $45.07 \pm 0.11$ | 1.65 |
| 7 | $1.60 \pm 0.22$ | $44.56 \pm 0.33$ | 1.64 |
| pH | Permanganate/Mn$^{2+}$ mol ratio 2/3 | | |
| 3 | $12.79 \pm 1.29$ | $91.08 \pm 0.27$ | 2.35 |
| 5 | $4.61 \pm 1.58$ | $91.96 \pm 1.19$ | 2.43 |
| 7 | $5.43 \pm 1.94$ | $91.71 \pm 0.25$ | 2.41 |
| pH | Permanganate/Mn$^{2+}$ mol ratio 4/3 | | |
| 3 | $7.29 \pm 0.54$ | $73.01 \pm 0.54$ | 1.90 |
| 5 | $7.37 \pm 0.98$ | $79.47 \pm 0.76$ | 2.00 |
| 7 | $7.17 \pm 0.67$ | $82.19 \pm 0.63$ | 2.07 |

*3.5. Ni Recovery from Raffinate*

3.5.1. Nickel Extraction Using LIX® 84-I

Ni was isolated from raffinate using LIX® 84-I at pH 5. The effect of extractant concentration on Ni and Li is depicted in Figure 8. Increasing LIX® 84-I concentration did not affect Li extraction, while the minimum extractant concentration for complete removal of Ni from raffinate was 10% at single extraction test, A/O ratio of 1. At this condition, the log separation factor of Ni-Li was 4.8. By varying the A/O ratio (1–7), the McCabe-Thiele diagram is generated (Figure 9), which shows the maximum loading capacity of LIX® 84-I 10% was 0.9 g/L Ni. The theoretical stages required for complete Ni extraction at different A/O ratios are provided in Table 10.

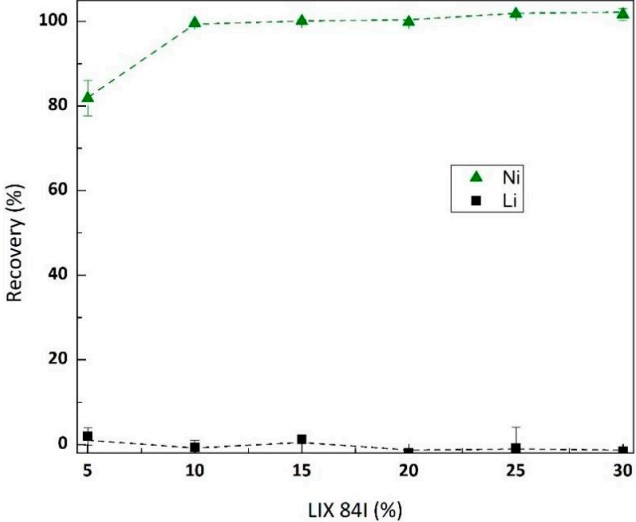

**Figure 8.** Effect of LIX® 84-I concentration (*v/v*) on Ni and Li extraction from raffinate (pH 5, metal concentration in feed Ni 0.13 g/L and Li 0.55 g/L).

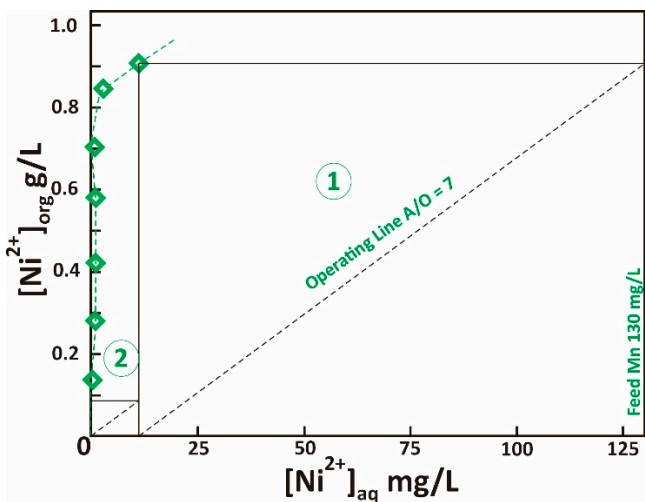

**Figure 9.** Extraction isotherm of Ni using LIX® 84-I 10% *v/v*, pH 5, McCabe-Thiele plot is shown for A/O ratio 7.

**Table 10.** Ni extraction rate and theoretical stages required for complete extraction at different A/O ratios (LIX® 84-I 10% *v/v*, feed solution 0.13 g/L, pH 5).

| A/O Ratio *v/v* | Ni Rec (%) | Number of Theoretical Stages |
|---|---|---|
| 1 | 99.79 ± 0.09 | 1 |
| 2 | 99.26 ± 0.01 | 1 |
| 3 | 99.26 ± 0.01 | 1 |
| 4 | 99.28 ± 0.04 | 1 |
| 5 | 99.41 ± 0.20 | 1 |
| 6 | 97.96 ± 1.45 | 2 |
| 7 | 91.69 ± 1.08 | 2 |

### 3.5.2. Stripping of Ni

As in the Cu stripping study, three mineral acids were tested in Ni recovery from loaded organic (LIX® 84- I 10% *v/v*, Ni 4.2 g/L). The effect of acid type and concentration on the stripping rate is shown in Table 11. The stripping study showed that Ni could not be removed completely from loaded organic using mineral acid, even when the concentration reached 5 M. Ni stripping rate was not varied much, based on the acid type and concentration. In the case of sulfuric acid, the stripping rate of Ni decreased slightly when acid concentration increased. This asserted the effect of increasing sulfate ion concentration, which could depress nickel sulfate solubility in strip solution due to similar ion effect.

**Table 11.** Ni stripping recovery according to mineral acid type and acid concentration (O/A 1, feed organic LIX® 84-I 10% *v/v*, Ni 4.2 g/L).

| Acid Concentration (M) | Ni Stripping Recovery (%) | | |
|---|---|---|---|
| | Sulfuric Acid | Nitric Acid | Hydrochloric Acid |
| 0.1 | 61.18 ± 0.24 | 50.84 ± 3.44 | 56.32 ± 0.15 |
| 0.5 | 62.44 ± 0.70 | 62.87 ± 0.28 | 55.87 ± 2.06 |
| 1 | 62.26 ± 1.18 | 65.79 ± 0.51 | 62.22 ± 1.17 |
| 2 | 61.01 ± 0.35 | 65.86 ± 0.34 | 60.07 ± 0.87 |
| 5 | 58.19 ± 1.42 | 63.50 ± 1.18 | 63.93 ± 4.31 |

### 3.6. Proposed Sequential Isolation of Transition Metals from Tannic Acid-Acetic Acid Media

Based on extraction and stripping data provided in previous sections, a flowsheet for sequential isolation of Cu, Co, Mn and Ni from leachate as a product of LIB black mass

leaching using tannic acid- acetic acid as lixiviant is proposed (Figure 10). In the stripping step, sulfuric acid is chosen considering the cost compared to the other two mineral acids. Also, sulfuric acid possesses less oxidation power than nitric and hydrochloric acid, which is deemed beneficial for preventing extractant degradation and corrosion to the extraction installation. Aside from solvent extractions, the flowsheet also contains precipitation steps for Co-Mn separation. Among three precipitants, i.e., oxalic acid, sodium hypochlorite and potassium permanganate, sodium hypochlorite is considered the best precipitant, based on precipitation selectivity values.

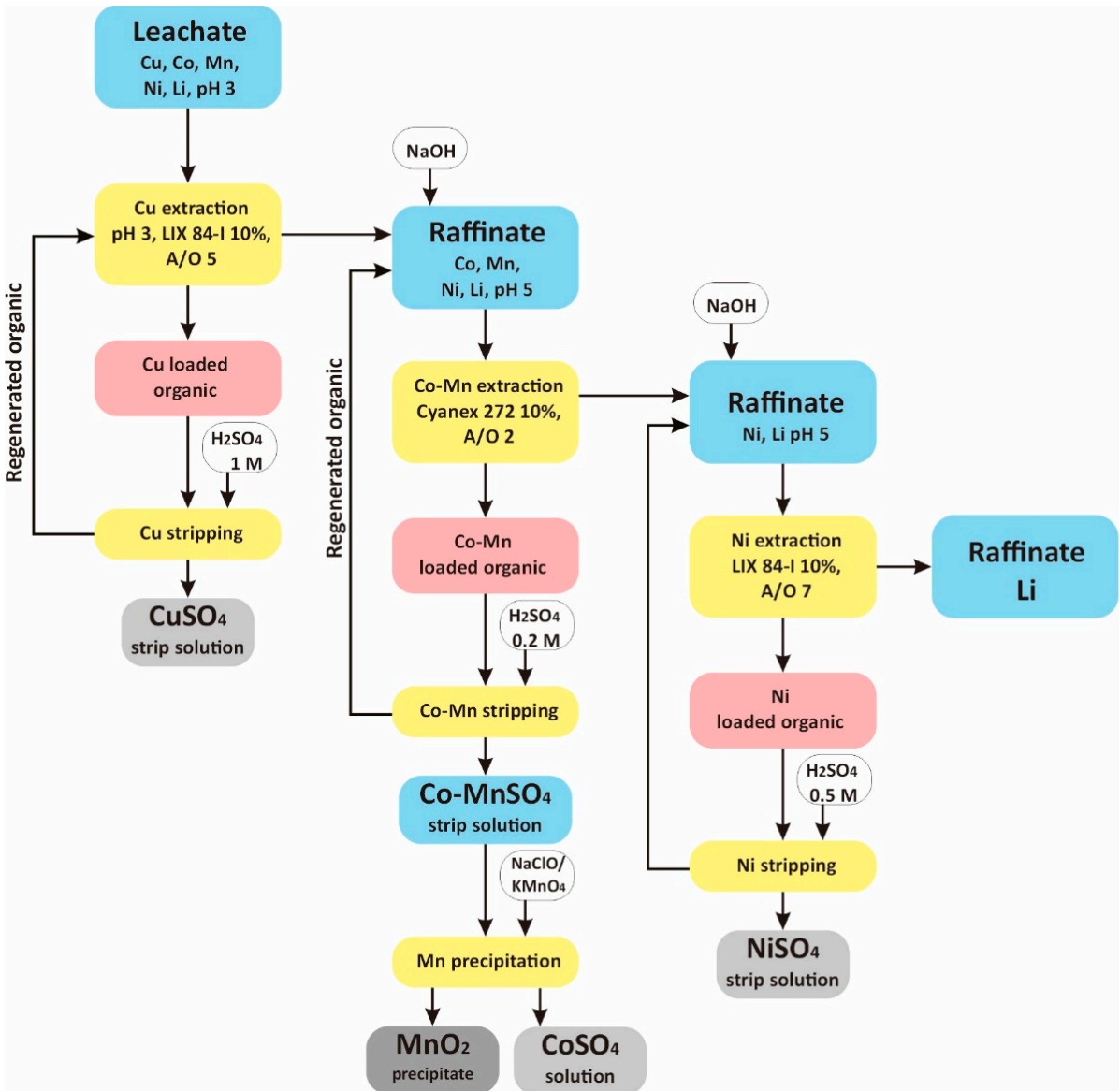

**Figure 10.** Proposed scheme for transition metal fractionation from LIB leach liquor (tannic acid-acetic acid media) by solvent extraction and oxidative precipitation.

## 4. Conclusions

Fractionation of transition metals (Cu, Co, Mn and Ni) from leachate produced from lithium battery black mass leaching using tannic acid-acetic acid was carried out. Each metal was successfully isolated using LIX® 84-I and Cyanex® 272-kerosene solvent extraction and permanganate or hypochlorite precipitation system. The pH of the leachate dictates the separation scheme of the metals, which follows the sequence: Cu separation (LIX® 84-I, pH 3), Co-Mn co-extraction (Cyanex® 272, pH 5) and Ni extraction (LIX® 84-I, pH 5) and

finally Li-rich solution. The third phase was not observed during the extraction, and sulfuric acid proved to be suitable for organic phase regeneration, concentrations of which varied: 1 M (Cu stripping), 0.5 M (Ni stripping) and 0.2 M (Co-Mn stripping). Co-Mn separation from strip solution was achieved by selective oxidation of $Mn^{2+}$ as $MnO_2$ precipitate using hypochlorite or permanganate as precipitant.

**Supplementary Materials:** The following supporting information can be downloaded at: https://www.mdpi.com/article/10.3390/met12050882/s1, Figure S1. Plot between equilibrium pH and metal recovery using Cyanex® 272 and LIX® 84-I (extractant concentration 10% *v/v* in kerosene, A/O 1), Figure S2. Cu, Co, Mn, Ni speciation in acetate media; Figure S3. FT-IR; Figure S4. Hydra and Medusa modelling for species distribution of Co and Mn in oxalate media.

**Author Contributions:** Conceptualization E.P.; methodology E.P., A.F.J. and W.A.M.; software M.A.M.; validation E.P., W.A.M. and M.A.M.; formal analysis E.P., C.A., A.F.J. and W.A.M.; investigation E.P., A.F.J., W.A.M., M.A. and F.B.; resources E.P., C.A., A.S.H., M.A., M.A.M. and F.B.; data curation E.P. and W.A.M.; writing—original draft preparation E.P., A.F.J.; writing—review and editing E.P., C.A., A.F.J., W.A.M., A.S.H., M.A., M.A.M. and F.B.; visualization E.P., A.F.J. and M.A.M.; supervision E.P., W.A.M., M.A. and F.B.; project administration E.P., A.S.H., F.B. and M.A.; funding acquisition E.P., C.A. and A.S.H. All authors have read and agreed to the published version of the manuscript.

**Funding:** This research was funded and supported by Indonesian National Research and Innovation Agency, Rumah Program Riset Material Maju FY 2021–2022. Open access publication was supported by Kroll Institute for Extractive Metallurgy, Colorado School of Mines.

**Institutional Review Board Statement:** Not applicable.

**Informed Consent Statement:** Not applicable.

**Data Availability Statement:** Not applicable.

**Conflicts of Interest:** The authors declare no conflict of interest.

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
