# Peer review of "Fractionation of Transition Metals by Solvent Extraction and Precipitation from Tannic Acid-Acetic Acid Leachate as a Product of Lithium-Ion Battery Leaching"

_metals, doi:10.3390/met12050882_

Round 1
Reviewer 1 Report
This paper proposes a method to separate copper, cobalt, manganese, and nickel from tannic acid-acetic acid leach liquor of lithium-ion battery using solvent extraction and precipitation. The development of recycling technology of lithium-ion battery is an important issue, and the proposed method seems to be new and effective. Thus, this paper is essentially publishable in the journal. But major revision is necessary to meet the academic quality as pointed out below.
- L21: “(followed by cobalt“ should be “followed by cobalt“.
- L55: “which concentration within mg/L order” should be “of which concentrations are within mg/L order”
- L78, “Our previous study”: Please cite the appropriate literature.
- L103, “in the previous study”: Please cite the appropriate literature.
- L114 and other lines: “ml” should be “mL”.
- L124: “separation factor” should be “log separation factor”
- L125: “Equation” should be “Equations”.
- L126: “Equation (5)” should be “Equation (4)”
- L127: Zero in “C0” should be subscript.
- L142: “which precipitant” should be “in which precipitant”.
- L146: “are” should be “were”.
- L162: Does the “pH” indicate pH at equilibrium, pH after extraction, or initial pH?
- L163, “Li was not virtually extracted”: Just before this clause, the authors write that the maximum extraction rate for Ni was 24%. According to Fig. 1a, the extraction behaviors of Ni and Li with Cyanex 272 are similar each other. Thus, they cannot claim that Li was not extracted”.
- L164: The authors write that pH50 for Mn, Co and Cu are 4.5, 4, 3.75, respectively. But, according to Fig. 1a, these are about 3.6, 3.9 and 4.2, respectively.
- L178: “Cu(CH3COO)3” should be “Cu(CH3COO)3-”(minus is superscript).
- L179-L181: The authors should show the extraction behavior of each metal from sulfate solution for comparison. Otherwise, the readers cannot understand how the behavior was changed in acetate solution. Also, what is important in the extraction behavior in the presence of aqueous complexing agents is the free metal concentrations, because it is common to consider that only a free metal ion is extracted. Thus, they should discuss the difference in the change of extraction behavior of each metal in terms of the change in free metal concentration rather than the kinds of metal-acetate complexes.
- L185-L187: Please cite the literature. Is this a comparison with a sulfate system?
- L191: If we discuss at the same initial pH, it would be true. But, if we discuss at the same equilibrium pH, the buffering effect has no relation.
- L204: “732.38” should be “732”. Please consider the significant numbers.
- Table 3: There is a plus or minus. This would mean that they wanted to express the standard deviation. So, how much was n (number of repetition)? In the figures also, there are error bars. Please explain the meaning of them and how to determine them.
- 3: The McCabe-Thielle plot in this figure is for cross-current extraction using fresh organic phase in each stage, although counter-current extraction is usually applied. Is there any special reason for accepting cross-current extraction? It is strongly recommended to study counter-current extraction. If they insist on cross-current extraction, they should state that this diagram is for cross-current extraction and explain the reason why they accepted it. Also, since the cross-current extraction is easy to carry out in batch experiment, it is strongly desired that they validate the results by batch experiment.
- 5: Please draw the axes and the explanation of the axes in black.
- 5: The same as Fig. 3. The numberings are in error. In Fig. 5b, the diagram is for combined flow (cross-current and counter-current extraction). It is very confusing.
- L275-L276: Did they observe the precipitation of metal sulfates?
- 7: Please draw the axes and the explanation of the axes in black.
- 7: The same as Fig. 3. In Fig. 7a, the diagram is for combined flow (cross-current using fresh aqueous phase for each stage and counter-current extraction).
- L294: “which” should be “in which”.
- L323: “3.4.1” should be “3.4.2”.
- (7)-(13): It seems to be common to express the standard oxidation-reduction potentials by E0 (0 is superscript) not by V.
- L349: “3.4.1” should be “3.4.3”.
- L370: “extraction” should be “extractant”.
- 9: The same as Fig. 7.
- L387-L388: It is known that extraction of nickel and its stripping with sulfuric acid are slow. I am afraid that stirring for 15 min (L117) was not sufficient for equilibration.
- L392: Did they observe the precipitation of nickel sulfate?
- Table 11: Use of high concentration of nitric acid is not recommended for LIX84-I because it would be degraded.
- 10: The final purpose of this study to propose the separation flowsheet for industrial scale operation. It is recommended to draw the flow sheet not using illustration of glass wares but using block diagram.
Author Response
The reply attached.

Reviewer 2 Report
This paper is focused on the development of a hydrometallurgical process to recover cobalt, nickel, manganese, lithium and copper from a leach solution produced by blackmass digestion with tannic acid-acetic acid. This paper addressed a hot topic and focused on the extraction and separation of the above-mentioned metals from organic media by solvent extraction combined with precipitation. Only several studies reported solvent extraction of these metals from tannic acid-acetic acid medium whereas more studies reported the use of oxidative precipitation to separate cobalt and manganese, which were previously co-extraction by solvent extraction.
The paper is well-written, and the papers brings interesting information for the community. Therefore, this paper is suitable for publication provided that the authors take into account the following minor comments:
- Page 4, first paragraph: please, define give the meaning of beta (separation factor); write C0 with the subscript ‘0’
- Page 4, last line and the rest of the manuscript: Please, indicate in the manuscript if the percentage are vol., wt or mol. Percentage (for instance ‘20%’ in the last line of Page 4.
- The extraction curves like in Fig. 1 are plotted as a function of the initial pH according to the authors (see figure caption). Usually, extraction curves are plotted as a function of the pH at equilibrium. Please, replace initial pH by pH at equilibrium in the whole manuscript.
- Page 5: the discrepancies between this work and other works reported in the literature mentioned in page 5 could be explained by the fact that the authors compare initial pH from their study to pH in other works, which may be pH at equilibrium. Please, could the authors check?
- In the manuscript, please replace ‘Mc Cabe-Thielle’ but ‘Mc-Cabe-Thiele’
- The figures in the supporting information are not readable. Please, check.
Author Response
The response attached.

Reviewer 3 Report
The present investigation focuses on the copper, cobalt, manganese and nickel recovery and separation by SX and precipitation processes, from leachate, produced from spent lithium-ion battery leaching using tannic acid and acetic acid as lixiviant.
It represents a nice technical contribution, having objectives and work methodology clearly established. It is clearly written and organized. The grammar and sentence structure are fine. In my opinion, it meets the publishing standards of “Metals” and could be published after minor revision.
Comments
1. concentration of each metal in leach liquor is listed in Table 2.
Full chemical analysis of the leach liquor should be given
2. Extraction test was carried out by batch method, by introducing typically 10 ml leach liquor and 10 ml organic phase
The volumes are too small (problems in phase separation?)…
3. Extraction recovery (R, %), distribution coefficient (D) and separation
Equations 1-4 are fundamental knowledge and could be omitted
4. The results of isotherm studies are depicted in Figure 1.
This is not an isotherm study but the %extraction vs pH. Isotherm curves study has to do with the increase of volume phases ratio, in constant conditions, which the Authors have presented for all scenarios.
5. Based on isotherm studies, sequential isolation of transition metal from tannic acid-193 acetic acid leach liquor is proposed as follows
A table with the metal concentration (before and after) values would be very helpful.
6. The Loaded organic (Cu content 4.8 g/L) was stripped at O/A volume
Why 4.8 g/L?
Initial Cu concentration in the leach liquor was 0.25 g/L.
With A/O=4 the maximum concentration of Cu would be 1.0g/L.
Author Response
The response attached.

Round 2
Reviewer 1 Report
The authors should reconsider the McCabe-Thiele analyses. Please read the details below.
(1) The authors changed ‘McCabe-Thielle’ to ‘Mc-Cabe-Thielle’ by following the other reviewer’s suggestion (that reviewer suggested ‘Mc-Cabe-Thiele’). But ‘McCabe-Thiele’ is the correct expression. Please check it.
(2) In the McCabe-Thiele analyses, I have understood that the authors assume cross current flow. On this assumption, there are serious mistakes in their analyses.
(2)-1 Fig. 5 a: Numbering is not in the right place. ‘2’, ‘3’, and ‘4’ in the figure should be ‘1’, ‘2’, and ‘3’, respectively. ‘1’ in the figure should be deleted.
(2)-2 Fig. 5 a: The location of operating line for the second extraction (number 3 in the current figure) is in error. The line should pass through ([Co]aq, [Co]org) = (0.5 g/L, 0) with a slope of 2. The current operating line means that the second extraction is done using the organic phase containing 0.5 g/L Co.
(2)-3 Fig. 5 a: In the third extraction (number 4 in the current figure), vertical line connecting the equilibrium point and the starting point of the operating line should be drawn.
(2)-4 Fig. 5 b: Like (2)-1, numbering is not in the right place.
(2)-5 Fig. 5 b: The second to fourth extraction (number 3, 4, and 5 in the current figure) is NOT the drawing for cross current flow but that for counter current flow.
(2)-6 Fig. 7a: The second to fourth stripping is NOT the drawing for cross current flow but that for counter current flow.
(2)-7 Fig. 7b: The location of operating line for the second stripping is in error. The line should pass through ([Mn]org, [Mn]aq) = (about 5 mg/L, 0) with a slope of 4. Current operating line means that the second stripping is done using the aqueous phase containing about 0.2 g/L Mn.
(2)-8 Tables 5 and 6: Because of these serious errors, the percentage recovery in these tables do not seem to be reliable.
(2)-9 McCabe-Thiele analysis is basically for the extraction or stripping of single component and not suitable for co-extraction or co-stripping system like in these cases (Co and Mn). It may be better to delete the McCabe-Thiele analyses in Figs. 5 and 7 and to show only the experimental results of multistage cross flow treatment in a few cases. If they cannot carry out these further experiments, they could do without them. Nevertheless, if they show the McCabe-Thiele analyses in Figs. 5 and 7, at least, the validation by the experiment is necessary.
(3) L181: ‘which complexes’ should be ‘of which complexes’.
Round 3
Reviewer 1 Report
I have found that the figures are appropriately corrected and have no further comment.